# OpenReview forum: "Truth is Universal: Robust Detection of Lies in LLMs"
_NeurIPS.cc/2024/Conference — NeurIPS 2024 poster_

### Official Review · Reviewer_Lo3X · 2024-07-05

**Soundness:** 2
**Presentation:** 2
**Contribution:** 3
**Rating:** 7
**Confidence:** 4

**Summary:**

This paper provides the analysis of linear subspaces of activations in LLMs in order to detect the truthfulness of the answer. Authors show that: 1) there are at least two-dimensional subspace for two types of false statements (affirmative and negated statements), which is the reason why previous approaches generalized poorly; 2) these two dimensions appears in various LLMs (Gemma-7B, LLaMA2-13B and LLaMA3-8B); 3) the detector built upon general truth direction is robust and it outperforms previous method (CSS)

**Strengths:**

1. Authors provide novel insight – identification of a two-dimensional subspace comprising a general truth direction and a polarity-sensitive truth direction is a significant contribution
2. Comprehensive evaluation including many existing datasets and LLMs.
3. The method achieves high accuracy in detecting both simple and complex lies, outperforming previous approach (CSS) by a notable margin.
4. Useful analysis of these directions using evaluation and principal components

**Weaknesses:**

1. Analysis lacks theoretical background of the reason for these directions to emerge in LLMs
2. Authors didn't provide any comparison with similar classifier-based like ITI [1] in order to mitigate hallucinations by intervention. In other words, it is not clear whether this general truth direction could be used to make model more honest.
3. The examples of failures of classifier are not provided, but could be important

[1] Li, Kenneth, et al. "Inference-time intervention: Eliciting truthful answers from a language model." Advances in Neural Information Processing Systems 36 (2024).

**Questions:**

My main questions are as follows:
1. Could you provide the analysis and examples of facts when the developed method fails? It might contain interesting insights on the possible improvements
2. How can it be that adding cities_conj spoils facts_disj in the Fig. 5? If it is possible, could you provide the accuracy matrix for cross-domain generalization to understand the dependence on domain, i.e what accuracy will be if we train on cities + neg_cities and test on facts + neg_facts.
3. Could you please explain the experiment with scenarios? You have the following scheme: Context + Buyer: 'text' + Agent 'text'. Do you pass this text with or without Context in order to classify a lie?

Here are also some small comments and typos:

Line 141: Please, specify directly what j index means

Line 157 skipped linebreak

Eq (7) Shouldn't the factor before sum be 2/n_i ?

Line 282 please refer to the figure instead the «colored boxes below»

**Limitations:**

1. The main paper limitation, in my opinion, is that  authors provide evidence that this subspace is **at least** two-dimensional, limiting the results to affirmative and negated statements and its logical combinations. However, authors understand and mention it in Discussion section. Thus I find the claim that the truth is universal in the beginning (and in paper name) is a bit misleading.
2. Authors provide the investigation of only one type of embeddings – layer-wise embedding of the last token.

---

> ### Author Rebuttal · Authors · 2024-08-05
>
> Thank you for the thoughtful review. We are glad that you found our identification of the 2D truth subspace to be a significant contribution and appreciate your constructive feedback.
>
> Regarding theoretical background: We agree that a theoretical explanation for why these truth directions emerge inside LLMs would be highly valuable. However, this is significantly beyond the scope of our empirical work. Such an explanation might require a detailed analysis of the training dynamics of the LLM and/or significant theoretical breakthroughs.
>
> “Comparison with similar classifier-based like ITI [1] in order to mitigate hallucinations by intervention.” We agree that making models more honest through causal interventions based on the 2D truth subspace is an interesting question. However, we believe that  it is beyond the scope of this paper. We will mention in the revision that this is an exciting direction for future research.
>
> Regarding the examples of failures of the classifier: This is a good point. In reply, we analysed the LR classifier failures for several datasets. We observed two main failure modes for misclassified statements: In the first failure mode, almost all misclassified statements in a specific dataset had the same truth label. While the learned truth direction is still able to separate true vs. false statements, the reason for these errors is that the bias (learned from other datasets) did not generalize sufficiently well. In the second failure mode, the learned truth direction was not able to accurately separate true vs. false statements. We conjecture that a possible reason is that for these statements the LLM may not have been certain if the statement is true or false. This failure mode occurred in the inventors and in the scientific facts datasets. We will add a discussion of this in the revision, along with a few example sentences which were misclassified.
>
> “How can it be that adding cities_conj spoils facts_disj in Fig. 5?” This is a good question. We currently do not have a satisfactory explanation for this behaviour. In the revision, we will add the cross-domain generalization accuracy matrix, and make a more detailed study of this issue.
>
> “Could you please explain the experiment with scenarios?” Thanks for pointing out that this part was not clear enough. In the revision we will provide a more detailed and clear explanation. After generating the LLM response, we pass the text with the context to the LLM (Context + Buyer: 'text' + Agent 'text') and record the activations over the final token. The truth or falsity of the LLM’s response is then predicted by the classifier probes based on these activations.
>
> Regarding the small comments and typos: Thanks for catching these! We will correct/update the paper.
>
> “I find the claim that the truth is universal in the beginning (and in paper name) is a bit misleading.” Thank you for raising this issue. The claim of universality made in our paper is that all considered LLMs (including also the added Gemma-2-27B-Instruct), represent truth in a similar manner, for affirmative sentences, negated sentences and conjunctions thereof. We agree that the truth representations might differ in other yet unknown truth dimensions and will make this clear in the revision.
>
> “Authors provide the investigation of only one type of embeddings” Thanks for raising this important point which was not sufficiently clearly explained in the original submission. We choose the last token and a specific layer for a good reason: Marks and Tegmark [2023] showed empirically that truth information is encoded in the intermediate layers of the LLM after processing this last token, which encodes the end of the input statement. We will clarify this in the revision. The choice of layer is justified by the results shown in Figure 2.

---

> > ### Comment · Reviewer_Lo3X · 2024-08-08
> >
> > Thank you for the detailed answer, and for the new experiments regarding Mass Mean (MM). However, I still have unanswered questions about the study:
> > 1) Could you provide now some experiments on cross-domain generalization accuracy, at least some part of domains? I believe it should be feasible for such timing.
> > 2) I have some concerns regarding the experiment with scenarios. If you record embeddings (with context passed to LLM as well question and answer) and train LR classifier on them, I don't see any significance in results, the classifier just detects if the context asked to lie or not. Could you clarify the reason why you included this experiment and what should reader learn from it?
> >
> > I am willing to increase my score, if you address these points and there will not be any flaws.

---

> > > ### Author Response · Authors · 2024-08-08
> > >
> > > Thanks for getting back to us!
> > >
> > > Regarding experiments on cross-domain generalization: Thank you for raising this issue again. Here is the cross-domain generalization matrix:
> > >
> > > |                | cities | neg\_cities | cities+neg\_cities | cities\_conj | cities\_disj |
> > > |----------------|--------|-------------|--------------------|--------------|--------------|
> > > | cities         | 1.00   | 0.64        | 1.00               | 0.90         | 0.82         |
> > > | neg\_cities    | 0.50   | 1.00        | 1.00               | 0.61         | 0.81         |
> > > | facts          | 0.73   | 0.78        | 0.90               | 0.86         | 0.73         |
> > > | neg\_facts     | 0.51   | 0.86        | 0.78               | 0.60         | 0.64         |
> > > | facts\_conj    | 0.54   | 0.53        | 0.71               | 0.71         | 0.58         |
> > > | facts\_disj    | 0.54   | 0.57        | 0.64               | 0.54         | 0.56         |
> > >
> > > The columns of this matrix correspond to different training datasets and the rows to different test sets. For example, the first column shows the accuracies of a LR probe trained on the cities dataset and tested on the six test sets. We train all LR probes on 80% of the data, evaluating on the held-out 20% if the test and train sets are the same, or on the full test set otherwise. While we provide this matrix in Markdown format due to the limitations of rebuttal comments, we will include a figure similar to Figure 5 in the revision. If you have any other specific cross-domain generalization accuracies you would like to see, please let us know!
> > >
> > > Regarding the experiment with the scenarios: Thank you for raising this important point which was not clear in the original submission. First, note that we are not training the classifier probes on the real-world scenarios but only use them as a test set. We will clarify in the revision that the probes are trained only on the activations of the simple affirmative statements in Table 1 and their negations.
> > > Second, we agree that there is a risk that the probes detect the incentive to lie rather than the lie itself. In reply to this concern, we have now recorded scenarios where LLaMA3-8B-Instruct provides honest answers even when there is an incentive to lie and applied our lie detector probes to these scenarios. If the probes detected only the incentive to lie and not the lie itself, we would expect lie detection accuracies below 50% on these scenarios. However, the detection accuracies were 90 $\pm$ 11% (new method), 77 $\pm$ 22% (CCS) and 62 $\pm$ 17% (LR), indicating that the probes indeed detect the lie itself.
> > > In summary, this is a proof of concept that classifier probes trained on simple statements can generalise to lie detection in more complex scenarios. We will clarify this and include these results in the revision.

---

> > > > ### Comment · Reviewer_Lo3X · 2024-08-08
> > > >
> > > > Thank you for the immediate experiment and results, as well as the clearer explanation. I raised my score.
> > > >
> > > > Firstly, I see now the cross-domain generalization, and indeed it is observed that transfer from "facts" to "cities" is generally worse. I don't think it makes the method worse, but it's worth looking into. There is also the question of whether there are two domains such that they will have completely opposite $t_G$, which I believe is out of the scope of this paper, but I would still be interested in your findings.
> > > >
> > > > Secondly, I understand now the experiment with the scenarios, and in such setting it is indeed interesting. The supporting experiments with "incentive to lie and not the lie itself" makes it more convincing and strong.
> > > >
> > > > Best wishes for your paper, and let me know if you would like to discuss further.

---

> > > > > ### Author Response · Authors · 2024-08-09
> > > > >
> > > > > Thank you very much. There is no need for further discussion on our part.

---

### Official Review · Reviewer_bydL · 2024-07-11

**Soundness:** 3
**Presentation:** 3
**Contribution:** 4
**Rating:** 7
**Confidence:** 3

**Summary:**

The paper presents a discovery of truth vectors, specifically a general one and a polarized one, present in Large Language Models (LLMs) when “lying”. The paper builds upon previous work by using vectors from intermediate-layer vector presentations to find these two vectors. This paper finds that one needs two truth vectors to account for negations and that doing so can also account for conjunctions between statements for determining lying by the LLM. These vector results generalize across a number of topics and models. These vectors can further be used in simple linear models to predict, with reasonable accuracy, whether an LLM is lying.

**Strengths:**

The paper is strong in its validation, novelty, and significance. For its significance, understanding LLM behavior, particularly undesirable behaviors, is very significant for the positive use of these models. This paper is directly addressing one of those behaviors. The paper also does a good job of building a robust dataset and running tests to show that the two truth vectors really do exist and are useful in lie detection. Finally, the discovery of not only the second truth vector, but its application for more universal truth detection is novel and will likely be of interest to the community (i.e., how universal is it? Are there other types of universal vectors? Etc.).

**Weaknesses:**

The paper has a few weaknesses in clarity. I do wish the code was released, as some of the descriptions of the tests are hard to follow as are how the linear models were constructed. For example, I am not quite sure what this means when getting the $a_{ij}$ vectors means, “we feed the LLM one statement at a time and extract the residual stream activations in a fixed layer over the final token of the input statement. The final token is always the period token (".").” Does this mean I am supposed to take the vector corresponding to the ”.” token? Also, could you give the actual names of the layers you extract this from in addition to a number, so that which layers these vectors are being extracted from is less ambiguous?

**Questions:**

-	How similar were $t_G$ and $t_P$ between the different LLMs? It's not clear to me from the main sections or Appendix B how similar, or universal $t_G$ was between the models. I wonder if because many open-source LLMs are trained on the same datasets if this is a result of the data used to train LLMs (especially the pre-training), and that that might be where the universality comes from.

-	What would happen if you asked LLM to evaluate whether it had produced a lie? Would that give a more accurate prediction of whether it lied in its previous output? Also, could something like this prompting scheme also be used with the intermediate layers method proposed in this work for possible performance improvements?

**Limitations:**

The authors have successfully addressed the limitations of the work. I especially appreciate that they were careful with not over-generalizing their results and indicating where this work stops and where future work could continue.

---

> ### Author Rebuttal · Authors · 2024-08-05
>
> Thank you for your positive review! We appreciate the criticism regarding clarity. In the revision, we will improve the writing throughout the manuscript.
>
> Regarding code release: We fully agree. In the spirit of reproducible research, with the revision, we will make our code and scripts public, so other researchers can reproduce the results of our manuscript.
>
> Regarding the extraction of the activation vectors: Thanks for raising this issue which was not sufficiently clear in our original submission. You are correct that we extracted the activation vector that corresponds to the “.” token at the end of the input statement. This final token contains the most enriched representation of the information present in this sentence. This scheme is identical to that of Marks and Tegmark.  We will  explain this in the revision.
>
> Regarding the names of the layers from which we extract the activations: Thank you for raising this issue. In the revision we will explain more clearly from which layer we extracted the activations and what the residual stream is. The residual stream layers have no specific names beyond their layer number.
>
> “How similar were tG and tP between the different LLMs?” Note that $t_G$ and $t_P$ for different LLMs come from layers that may have different dimensions and as such may not be directly comparable. The universality phenomena we uncover is that the various LLM models (including from Llama and Gemma families) all have a similar 2D truth representation.  We will clarify this in the revision.
>
> Note that Google DeepMind and Meta AI do not publish their training datasets. Hence, it may be plausible that the training datasets are very similar, but we do not know this for sure.
>
> Regarding asking the LLM whether it just lied: This is an excellent question. However, it has already been explored in previous work by Pacchiardi et al. [2023] , so we did not include it again in our paper. We will clarify this important point in the revision. Pacchiardi et al. [2023] asked GPT-3.5 "Are you lying?" after it generated a lie, and found that GPT-3.5 lied again more than 80% of the time (see Appendix B.2.2 of their paper). In general, our approach is designed for scenarios where the LLM knows it is lying, but is doing so in pursuit of some goal or reward, see for example Scheurer et al. [2023]. If it is a sufficiently capable LLM, it will not reveal that it has just lied, but will hide it. But from the internal model activations we might be able to detect the lie! We will add a discussion of this in the revision.
>
> “Also, could something like this prompting scheme also be used with the intermediate layers method proposed in this work for possible performance improvements?” This is an excellent suggestion for future research. We will mention this in the revision.

---

> > ### Comment · Reviewer_bydL · 2024-08-13
> > **Reply to Rebuttal**
> >
> > Thank you for addressing the clarity issues with the paper and for committing to releasing your code. I stand by my rating as an accept for the paper.

---

### Official Review · Reviewer_YJ6m · 2024-07-12

**Soundness:** 3
**Presentation:** 4
**Contribution:** 2
**Rating:** 6
**Confidence:** 3

**Summary:**

The authors study LLM lie detection using probes trained on model internals. They show that LLM representations contain a two-dimensional subspace that corresponds to a general truth direction as well as a polarity-dependent truth direction which is sensitive to whether the statement is affirmative or negated. This clarifies the observation of prior works which observed the lack of generalizability of lie detection probes, and also enables a classifier that outperforms Consistent Contrastive Search, an existing lie detection method.

**Strengths:**

- **Clarity and presentation**: the paper is easy to follow, well-structured and clearly presents the relevant context, experimental details and findings.
- **Explains prior observations regarding generalizability**: I found the 2D subspace explanation convincing, and the fact that the first and second principal components, corresponding to tg and tp, explain a large proportion of the variance to be a neat result
- **Universality**: The authors show that this subspace is present in multiple models, including Llama3-8B, Llama2-13B and Gemma-7B.
- The authors also look for the presence of additional linear structure/dimensionality by studying conjunctions, coming to the conclusion that it does not have a significant impact.

**Weaknesses:**

- **Novelty**: although, the paper's insight about a 2D subspace is interesting, much of this result relies on findings of prior work, such as Marks and Tegmark who show that training on statements and their opposites improves generalization, and Levinstein & Herrmann who observe the failure to generalize on negated statements. Additionally, this insight does not lead to any novel method of training a more generalizable truth probe.
- **Generalizability**: Truth Directions learnt by probes are still entangled with topic-specific features, as suggested by the fact that projecting out tg and tp from activations still leads to good performance on subsets of the city training data (Figure 5). Although training on a range of topics reduces this issue, it still seems unclear to me that a robust generalizing set of truth directions can be found, especially when tested on more challenging and out of distribution statements.  A more comprehensive investigation of generalizability would strengthen the paper.
- **Universality:** claims of universality would be strengthened by extending analysis to larger models and other model families.
- **Room for causal experiments**: Prior work (Marks and Tegmark) also investigates the casual effect of their discovered truth directions.

**Questions:**

-  Did you investigate the Mass Mean Probing technique proposed by Marks and Tegmark? If so, how does it compare?
- Having presented a method to obtain a disentangled truth direction tg in Figure 3, why do you choose to train a normal LR probe on balanced statements in Section 5 when testing Generalization? Do they find similar directions (I would assume so, since as you mention, the balance of statements discourages the learning of tp)? Is there a reason to expect one method to be superior to the other? Does the first and second principal component of this probe also correspond significantly with tg and tp?

**Limitations:**

Yes, the authors identify that their evaluation of generalizability remains limited and that their investigation could be extended to larger and multimodal models.

---

> ### Author Rebuttal · Authors · 2024-08-05
>
> Thanks a lot for your review! We are glad you found the 2D subspace explanation convincing and liked the presentation of the results.
>
> Regarding Novelty: We agree that our work was directly motivated by the empirical findings of Marks and Tegmark and Levinstein & Herrmann. The novelty and importance of our work is two-fold. (i) We explain their empirical findings via our 2D truth representation subspace analysis; and (ii) our analysis clarifies why linear classifiers may still be highly accurate for the task of truth/falsity classification. We will clarify this in the revision.
>
> “this insight does not lead to any novel method of training a more generalizable truth probe”
> As mentioned above, the insight that there is a 2D subspace explains why linear classifiers may still be highly accurate in classifying truth/false statements. In direct reply to the above concern, in recent follow up work (after our original submission), based on this insight, we constructed a new classifier, which is still linear, but achieves an even higher accuracy than LR and CCS. We will gladly discuss this in the revision.
>
> Regarding generalizability and testing on more challenging and out of distribution statements: Please note that Section 5 presents such settings. Specifically, to the best of our knowledge, we are the first to quantify the generalisation accuracy of lie detectors based on internal probe activation, when tested on a variety of challenging real-life scenarios. We would like to emphasise that Pacchiardi et al., the creators of the real-life scenarios, used the output of the LLM to follow up questions (after lying or telling the truth) and not the internal activations of the LLM to classify the LLM responses as truthful reply or lie. We will clarify this important point in the revision.
>
> “A more comprehensive investigation of generalizability would strengthen the paper”: In general, we agree that further investigations are of interest and we explicitly mentioned this in the original submission. That said, please note that the evaluation of generalizability in our submission is more comprehensive than most prior works, both in terms of type of statements, number of diverse datasets considered, etc. We will clarify this in the revision.
>
> Regarding Universality: We agree that this claim can be strengthened by considering larger models and other model families. In reply to your concern, we extended our analysis to Gemma-2-27B-Instruct, a model twice the size of the largest model considered in our original submission, and to Mistral-7B-Instruct-v0.3, a LLM from a different model family. The results are qualitatively the same as for the other LLMs with a 2D truth subspace in the activation space of the models. We will include these results in the appendix of the revision.
>
> Regarding causal experiments: We agree that causal intervention based on the 2D truth representation is an exciting research direction. However, it is beyond the scope of this paper, as our focus lies on providing rigorous evidence for the existence of the 2D truth subspace and using this knowledge for robust lie detection. We will mention in the revised conclusion section that causal intervention based on our insights is an exciting direction for future research.
>
> Regarding the Mass Mean Probing technique of Marks and Tegmark: We thank the referee for this suggestion. In response, we did a comparison with MM probing on the Llama3-8B-Instruct model. For a fair comparison we in fact extended the MM probe to include a learned bias term. The MM probe generalised a bit better than our original LR based lie detector classifier. However, a new classifier we constructed after our original submission (mentioned above) achieved even higher generalisation accuracies. We included a table comparing the four methods in the global response. We will include all of these results in the revision.
>
> Regarding the normal LR probe in Section 5: This is an excellent question and we agree that the motivation for this was not sufficiently well explained in the original submission. Our analysis of the 2D truth subspace showed that there is a general truth direction $t_G$ which we can disentangle from $t_P$ by training a linear classifier on a balanced number of affirmative and negated statements. Logistic Regression was simply our choice for the linear classifier and the direction it learned is indeed similar to $t_G$. We will clarify this in the revision.
>
> Is there a reason to expect one method to be superior to the other?
> Good question! Empirically, we have found (after our original submission) that truth directions which are learned separately from the bias (as in Section 3) generalise better to the unseen statement types and real-world scenarios than truth directions which are learned together with the bias (as in LR). However, we are not aware of any fundamental reason why one method would be superior to the other. We will discuss this in the revision.

---

> > ### Comment · Reviewer_YJ6m · 2024-08-11
> >
> > Thanks for your detailed response: I appreciate the additional experiments you conducted for MM probes as well as universality across models. I will keep my current score.

---

### Official Review · Reviewer_XXUN · 2024-07-15

**Soundness:** 2
**Presentation:** 2
**Contribution:** 1
**Rating:** 4
**Confidence:** 3

**Summary:**

The paper considers the problem of detecting whether a statement is true or false from the intermediate activations of an LLM. It starts by introducing a linear functional form with two linear components t_G and t_P which is able to discriminate between true and false statements for both affirmative and negated statements. They then show that most of the variance in the activations is captured by its first two principal components, which closely match t_G and t_P. They then show that removing the linear subspace spanned by t_G and t_P hurts the generalization performance of a linear probe (i.e., logistic regression over the activations) in discriminating between true and false statements. They then show that logistic regression with balanced affirmative and negated statements outperforms a previous approach called Contrast Consistent Search, and include results for 26 real-life role-playing scenarios.

**Strengths:**

The paper is clear. It directly addresses a problematic of prior work, namely constructing classifiers that are more robust in detecting truthful statements using LLMs’ intermediate activations. The findings of the work seem correct: training the classifiers on more diverse (and balanced) data improves generalization.

**Weaknesses:**

The analysis of Section 3 is qualitative and would benefit from including the classifiers’ accuracy (i.e., accuracy gain when including t_G). Looking at Figure 1 top right, it seems like a single hyperplane would be able to separate True/False with reasonably high accuracy. Therefore, I don’t see the value of the “truth direction” t_G and “polarity-sensitive truth direction” t_P decomposition proposed by the authors (other than for interpretability purposes), the important point seems to be to train on the negated statements. That is, to reduce the distribution shift from train to test.

For the analysis of Section 4, since the first two principal components account for most of the variance of the intermediate activations, and t_G and t_P are reasonably aligned with these two PCs, it seems unsurprising that performance would degrade after projecting out t_G and t_P — since one is removing most of the information contained by the activation functions. That is, I would expect generalization to degrade not only for the true/false discrimination tasks considered, but for other tasks unrelated to true/false discrimination. If removing the first two PCs yields qualitatively similar results, it seems that there is nothing too specific about “truth” being projected out, simply most of the information contained in the activations.

In Section 5, the authors compare against a prior method, CCS. I take the main results to be that a simple classifier trained on diverse and balanced data outperforms a more complex and specialized learning algorithm. More broadly, I believe that the main contribution of the paper is to show that, for the task of true/false discrimination from internal activation functions, training on more diverse and balanced data improves generalization.  I don’t think that this contribution is of sufficient significance to warrant a NeurIPs publication.

**Questions:**

I suggest the following:
* In Section 3, to include the performance of the logistic classifier trained on both affirmative and negated statements, and compare it to that of the proposed t_G and t_P decomposition.
* In Section 4, to compare to projecting out the first two principal components.

**Limitations:**

The authors adequately discuss the limitations of their work.

---

> ### Author Rebuttal · Authors · 2024-08-05
>
> Thank you very much for your review! We appreciate your comments and hope that our response below can convince you that the contribution of our work goes beyond just showing that training on more diverse and balanced data improves generalization.
>
> “The analysis of section 3 is qualitative”. Please note that the accuracies of the classifiers based on the directions $t_G$ and $t_P$ appear in Figure 3. This figure shows the clear advantage of classification based on $t_G$. The accuracy of a classifier trained only on affirmative statements ($t_A$), is 81%, whereas using $t_G$ it is 95%. We will add these numbers to figure 1 and to the main text. In addition, following the suggestion of the referee, we will add to Figure 3 another column with the accuracy of a logistic regression classifier trained on affirmative and negated statements.
>
> Regarding the importance of the two directions, the decomposition, etc: We agree with the referee that in general training on a larger dataset, more representative of future test instances, is often beneficial. However, it is a-priori unclear that a linear classifier would be suitable.
> In the following, we explain why in our opinion, our discovery of the 2D truth subspace spanned by $t_G$ and $t_P$ is a significant contribution. Prior to our work it was unclear how LLMs internally represent the truth or falsity of factual statements. For example, it was unclear whether there is a single "general truth direction" or multiple "narrow truth directions”, each for a different type of statement. However, this knowledge is essential in order to construct a robust LLM Lie Detector based on its internal model activations.
> In particular, insights from our 2D truth subspace representation can be directly used to construct robust truth classifiers, which are still linear. Furthermore, these insights might allow other researchers to construct even more accurate non-linear classifiers which leverage both dimensions of the truth subspace.
>
> Regarding the analysis of Section 4, and the first 2 PCs accounting for most of the variance: We thank the referee for raising this issue, which was not sufficiently clearly explained in the original submission. A key point is that we did not perform PCA on the raw activations. Instead, we first preprocess the data to isolate truth-related variance. As described in Section 4, we have a two-step process: (1) we center the activations in each dataset D_i, see L200-202. (2) We average the resulting activations for true and false statements  in each dataset, see Eq. 7. The first two PCs after this preprocessing capture 60% of the variance in the centered and averaged activations, but not in the raw activations. On the raw activations, these two vectors capture only ~10% of the variance. Therefore, we are not removing most of the information contained in the activation functions. We will clarify this important point in the revision.
>
> Regarding comparison to CCS, and the statement “I take the main results to be that a simple classifier trained on diverse and balanced data outperforms a more complex and specialized learning algorithm”:
> Here we respectfully disagree with the referee. First, let us point out that we trained both our method, as well as CCS on the same diverse and balanced training set. Second, the high accuracy of our method is supported by our 2D analysis of the truth representation in the activation space, hence showcasing the tight connections between the various sections of our manuscript.
>
> To conclude, given the clarifications above, as well as our replies to the concerns raised by the other reviewers, in our opinion, the contributions of our manuscript are of interest to the community and of sufficient significance to warrant publication at NeurIPS.

---

> ### Comment · Reviewer_XXUN · 2024-08-12
>
> Thank you for your response, and apologies for the delay in responding.
>
> > However, it is a-priori unclear that a linear classifier would be suitable.
>
> What is the significance of the classifier being linear? Do you assume that linear implies more robust, at little to no cost in accuracy? This might not hold beyond the toy tasks considered in this work (e.g., the method already fails to generalize to logical conjunctions well, let alone actually detecting “lies” in deployed systems).
>
> > Insights from our 2D truth subspace representation can be directly used to construct robust truth classifiers, which are still linear. Furthermore, these insights might allow other researchers to construct even more accurate non-linear classifiers which leverage both dimensions of the truth subspace.
>
> I fail to see what insights you used to construct the final linear classifier. Is it not just standard linear regression over the activations? What insights could be used to construct more accurate non-linear classifiers?
>
> > On the raw activations, these two vectors capture only ~10% of the variance.
>
> This addresses my concern regarding most of the variance being projected out.
>
> > First, let us point out that we trained both our method, as well as CCS on the same diverse and balanced training set. Second, the high accuracy of our method is supported by our 2D analysis of the truth representation in the activation space, hence showcasing the tight connections between the various sections of our manuscript.
>
> Yes, I was commenting under the assumption that both CCS and your method are trained on the same data. The fact that linear regression outperforms CCS is to me more indicative of CCS being a poor method for this particular task rather than of the merits of linear regression. Linear regression (linear probing) is typically the baseline for classification tasks. For the more challenging tasks (Figure 6b, real-world scenarios), the error bars are so large that it is not even clear that the performance of LR and CCS are significantly different.
>
> My general assessment remains “I believe that the main contribution of the paper is to show that, for the task of true/false discrimination from internal activation functions, training on more diverse and balanced data improves generalization” I’ll add the following “The work shows that a linear classifier is sufficient to obtain high accuracy in simple true/false discrimination tasks”. I still do not think that these two contributions are of sufficient significance to warrant publication.

---

> > ### Author Response · Authors · 2024-08-13
> >
> > Thanks for getting back to us.
> >
> > “What is the significance of the classifier being linear?”
> > As mentioned in the introduction (line 46-50), growing evidence supports the hypothesis that LLMs might encode human-interpretable concepts as linear combinations of neurons, i.e. as directions in activation space. Our manuscript provides strong evidence that one of these concepts might be the truth, represented by the truth direction $t_G$. Given that accurately assigning truth labels to statements is highly labor-intensive and data is correspondingly scarce, we aimed for a classifier with a strong inductive bias towards a solution that we have reason to believe generalizes well. Hence, our choice of a linear classifier.
> >
> > Regarding insights used for the construction of the linear classifier: Our insight was that training a robust linear classifier which generalizes to both affirmative and negated statements requires disentangling $t_G$ from $t_P$. We achieved that by training on activations from an equal number of affirmative and negated statements.  Note that successful disentanglement of $t_G$ and $t_P$ is far more crucial than disentangling $t_G$ from some spuriously correlated feature that correlates with truth in the narrow training distribution but is mostly uncorrelated with truth beyond it. In contrast to such spuriously correlated features, $t_P$ is consistently anti-correlated with truth on negated statements (as opposed to merely being uncorrelated), which is even worse for generalization. We will clarify this point in our revision.
> >
> > “What insights could be used to construct more accurate non-linear classifiers?”
> > We showed that the truth direction $t_P$ points from false to true for affirmative statements and from true to false for negated statements. By estimating the statement polarity from the activations, one could flip the sign of $t_P$ for negated statements such that it points from false to true for both affirmative and negated statements. Now one could use both dimensions of the truth subspace, $t_G$ and $t_P$, for classification, not losing valuable information compared to using just one dimension. In the top right panel of Figure 1 you can see that such a non-linear approach would probably improve the accuracy of the classifier compared to a linear approach. Of course, this is just an example and people might come up with other approaches based on our analysis of how truth is represented in LLMs. We will mention this in the revision.

---

> > > ### Comment · Reviewer_XXUN · 2024-08-13
> > >
> > > Thank you for the detailed response. I remain unconvinced about the significance of the work. Since I do not have any concerns regarding its technical correctness, and other reviewers are positive about the work, I'll raise my score.

---

### Author Rebuttal · Authors · 2024-08-05

In response to reviewer YJ6m's suggestion, we expanded our comparison of classifiers in Section 5. In addition to Logistic Regression (LR) and Contrast Consistent Search (CCS), we now include Mass Mean (MM) probing by Marks and Tegmark [2023] and a new classifier we developed after the original submission. Importantly, the design of our new classifier is motivated by the structure of the 2D truth subspace identified in our work. Our results, summarised in Table 1 of the attached PDF, demonstrate that our new classifier generalises even better than previous methods. We will include these results in the revision, along with a detailed explanation.

---

### Decision · Program_Chairs · 2024-09-25

**Decision:**

Accept (poster)

**Comment:**

We thank the authors for their submission as well as thorough and constructive engagement throughout the rebuttal period.

After a careful review and discussion by the reviewers and AC, we recommend that this work be accepted for publication. The reviewers point to several strengths of this paper, including the importance and timeliness of the problem the paper tackles, novelty of insight, comprehensiveness and soundness of evaluation, and clarity of exposition. We expect this paper to be of interest to the conference attendees and may inspire interesting future work.

As part of the camera ready revision, we strongly encourage the authors to incorporate the detailed feedback from the reviewers, some of which they already started to address in the rebuttals. These include the more substantive points (e.g., from reviewer XXUN), as well as the lower-level details (e.g., regarding examples of failure of classifiers as noted by reviewer Lo3X).